



# Stochastic modeling of flow and conservative transport in three-dimensional discrete fracture networks

I-Hsien Lee[1,2], Chuen-Fa Ni[1,2], Fang-Pang Lin[3], Chi-Ping Lin[1], Chien-Chung Ke[4]

[1]Graduate Institute of Applied Geology, National Central University
[2]Center for Environmental Studies, National Central University
[3]Data Computing Division, National Center for High Performance Computing
[4]Sinotech Engineering Consultants Inc.

*Correspondence to*: Chuen-Fa Ni (nichuenfa@geo.ncu.edu.tw)

**Abstract.** This study presents the stochastic Monte Carlo simulation (MCS) to assess the uncertainty of flow and conservative transport in 3D discrete fracture networks (DFNs). The MCS modeling workflow involves a number of developed modules, including a DFN generator, a DFN mesh generator, and a finite element model for solving steady-state flow and conservative transport in 3D DFN realizations. The verification of the transport model relies on the comparison of transport solutions obtained from HYDROGEOCHEM model and an analytical model. Based on 500 DFN realizations in the MCS, the study assesses the effects of fracture intensities on the variation of equivalent hydraulic conductivity and the exhibited behaviors of concentration breakthrough curves (BTCs) in fractured networks. Results of the MCS show high variations in head and Darcy velocity near the specified head boundaries. There is no clear stationary region obtained for the head variance. However, the transition zones of nonstationarity for x-direction Darcy velocity is obvious and the length of the transition zone is found to be close to the value of the mean fracture diameter for the DFN realizations. The MCS for DFN transport indicates that a small sampling volume in DFNs can lead to relatively high values of mean BTCs and BTC variations.

## 1 Introduction

Successful characterizations of flow and contaminant transport in fractured geologic formations depend on adequate descriptions of complex geometrical structures, which comprise a wide variety of fractures and their connections (Ahmed et al., 2015; Pichot et al., 2012; Weng et al., 2014). The fracture characteristics can be quantified by using various statistical parameters, including the fracture orientation, length, shape, and permeability alongside the fracture intensity and connectivity (Bonnet et al., 2001; Botros et al., 2008; Bour et al., 2002; Koike et al., 2015; Stephens et al., 2015). These commonly used parameters represent fracture networks at sites of interest and bridge gaps between limited field observations and flow and transport implementations for site-specific issues.

Using the discrete fracture networks (DFN) approach to characterize the flow and transport in fractured media is a challenging task for practical applications. Intensive research over the past three decades has led to the development of



numerous models that are based on the DFN approach to model the flow or transport in fractured formations (Cacas et al., 1990a; Cacas et al., 1990b; de Dreuzy et al., 2013; Hyman et al., 2015a; Liu and Neretnieks, 2006; Long et al., 1985; Pichot et al., 2012; Xu and Dowd, 2010). Advanced 3D DFN approaches typically include procedures of fracture generation, DFN meshing, and flow and transport or particle tracking (de Dreuzy et al., 2013; Erhel et al., 2009; Hyman et al., 2014; Pichot et

al., 2012; Xu and Dowd, 2010; Zhang, 2015). Particle tracking algorithms are usually preferred to simulate DFN transport and have recently been widely implemented to evaluate the time resistance of contaminants for fractured formations (Hyman et al., 2015a; Hyman et al., 2015b; Makedonska et al., 2015; Painter et al., 2008; Stalgorova and Babadagli, 2015; Wang and Cardenas, 2015). The objective of the Lagrangian approach is to avoid numerical difficulties in solving the advection dispersion equation (ADE) in complex DFN domains. Such DFN transport models use the released particles to represent the

contaminant with a specified mass or concentration. Many previous studies have discussed the issues in treating particle movement in fracture networks (Hyman et al., 2015a; Hyman et al., 2015b; Johnson et al., 2006; Makedonska et al., 2015; Painter et al., 2008; Park et al., 2003; Wang and Cardenas, 2015; Zafarani and Detwiler, 2013).

Over the years, many studies have focused on developing flow and transport models and integrating DFN simulation workflows for 3D fracture networks (Hyman et al., 2014; Hyman et al., 2015a; Lee and Ni, 2015). Specifically, the DFN

transport was mainly modeled based on Lagrangian approaches such as particle tracking and random walk algorithms (e.g., Makedonska et al., 2015; Painter et al., 2008; Stalgorova and Babadagli, 2015; Wang and Cardenas, 2015) . Numerical solutions to the ADE based on the Eulerian approach have not been widely implemented because of computational issues, such as numerical dispersion and convergence in the model for complex fracture connections (Odling, 1997).

With the advantages of computational technologies, the stochastic modeling of flow and Eulerian-based transport in 3D

DFNs has become a feasible task. It is an important issue to quantify flow and transport uncertainties based on available DFN properties. The objectives of this study are to develop and implement numerical models for stochastic modeling of flow and conservative transport in 3D DFNs. The stochastic Monte Carlo simulation (MCS) is conducted to assess the flow and transport uncertainty induced by the 3D DFNs. In this study, we first assess the developed ADE model by comparing the solutions of simple porous fractures against those from the HYDROGEOCHEM finite element model (Yeh et al., 2004) and

the analytical model developed by Wexler (1992). Then, we use the MCS to evaluate the equivalent hydraulic conductivity for specified DFN statistical parameters. The collected flow and transport realizations enable the analyses of flow and transport uncertainties in the fractured simulation domain. The simulation results are expected to provide general insight into the evaluations of flow and transport uncertainty based on the available DFN geometrical properties.

## 2 Mathematical formulas and numerical models

In this study, the fractures in a DFN are considered to be porous media with impermeable surfaces that are connected to the formation matrix. The two impermeable surfaces of a fracture are considered to be two rough parallel plates that enable



fluids to pass through the fracture at a relatively high velocity (e.g., Kwicklis and Healy, 1993; Lee and Ni, 2015; Pruess and Tsang, 1990). The following presents the mathematical formulas, a brief description of the mesh generation, and the finite element models for simulating the 3D DFN flow and transport.

**2.1 Flow and transport equations**

5 The mathematical formulation for the DFN consists of flow and transport in a set of 2D porous fracture plates connected in a 3D domain. The coupling of flow and transport in porous media has been widely investigated in fields that are related to groundwater hydrology (Dagan, 1989; Hartley and Joyce, 2013; Yeh et al., 2004; Zheng and Bennett, 2002). Based on the concept of mass conservation and Darcy's law, the equations for solving the steady-state and depth-averaged hydraulic head for 2D porous fractures can be expressed as

$$\nabla \cdot [K(\mathbf{x})b(\mathbf{x})(\nabla h(\mathbf{x})] + Q(\mathbf{x}) = 0 , \tag{1}$$

subject to the boundary conditions:

$$h(\mathbf{x})\big|_{\Gamma_D} = h_D , \tag{2}$$

and

$$\big[K(\mathbf{x})\nabla h(\mathbf{x})\big] \cdot \mathbf{n}\big|_{\Gamma_N} = q_N , \tag{3}$$

where $h(\mathbf{x})$ is the hydraulic head, $K(\mathbf{x})$ is the hydraulic conductivity and $b(\mathbf{x})$ is the aperture for fractures. For saturated flow, the locations of the fracture has been taken into account in Eq.(1). We assumed that the flows in fractures were parallel to the fracture and the relatively small fracture apertures insignificantly influenced the vertical flow in the 2D fractures. The notation $Q(\mathbf{x})$ represents the sources or sinks applied to the 2D porous fractures. The Cartesian coordinate $\mathbf{x}$ ( $\mathbf{x} = 1$ and 2) represents the x- and y-directions in the horizontal modeling domain of the fractures. Moreover, $h_D$ represents the prescribed

head at the Dirichlet boundary $\Gamma_D$, and $q_N$ is the specific flux at the Neumann boundary $\Gamma_N$. The notation $\mathbf{n}$ is a unit vector that is normal to the boundary $\Gamma_N$. In this study, we consider that the DFNs are fully saturated. The z-coordinates for fractures represent the elevation heads, which have been considered in the calculations of hydraulic heads. Therefore, the solutions of hydraulic heads in 2D fractures in the 3D domain can be obtained from Eqs. (1) to (3).

Similar to the flow simulation, the depth-averaged conservative solute transport equation for saturated fractured porous

media is governed by the ADE and can be written as (e.g., Dagan, 1989; Ni et al., 2009; Zheng and Bennett, 2002):

$$\frac{\partial c(\mathbf{x},t)}{\partial t} = -v(\mathbf{x})\nabla c(\mathbf{x},t) + \nabla \cdot [\mathbf{D}_e(\mathbf{x})\nabla c(\mathbf{x},t)] + Q_c(\mathbf{x},t) , \tag{4}$$

subject to the initial and boundary conditions:

$$c(\mathbf{x},0)\big|_{\Omega} = c_0 , \tag{5}$$

$$c(\mathbf{x},t)\big|_{\Gamma_D} = c_D , \text{ and} \tag{6}$$



$$\left[\boldsymbol{D}_e(\mathbf{x})\nabla c(\mathbf{x},t)\right]\cdot\mathbf{n}\Big|_{\Gamma_N} = q_c \,, \tag{7}$$

where $c(\mathbf{x},t)$ is the volumetric solute concentration that is measured in the liquid phase and $Q_c(\mathbf{x},t)$ represents the rate where the volumetric solute concentration is injected (source) or extracted (sink) from the DFN. The notation $v(\mathbf{x}) = -K(\mathbf{x})\nabla h(\mathbf{x})/n(\mathbf{x})$ is the seepage velocity and $n(\mathbf{x})$ is the effective porosity in the porous fractures. Calculating the seepage velocities at nodes relies on the obtained hydraulic heads at element centers in the DFN (i.e., the solution of Eqs. (1) to (3)). This study used an improved approach proposed by Yeh (1981) to obtain the seepage velocity at nodes to consider the global mass conservation of flow in the simulation domain. In Eqs. (5) to (7) $c_0$ represents the initial concentration in the entire modeling domain $\Omega$, $c_D$ is the specified concentration at the Dirichlet boundary, and $q_c$ is the dispersive flux at the Neumann boundary. Moreover, $\boldsymbol{D}_e(\mathbf{x})$ in Eq. (4) is considered the macro-dispersion coefficient, which is evaluated based on the seepage velocity (Zheng and Bennett, 2002):

$$\boldsymbol{D}_e(\mathbf{x}) = \left[\alpha_L(\mathbf{x}) - \alpha_T(\mathbf{x})\right]\frac{v_i v_j}{\bar{v}} + \boldsymbol{D}_0(\mathbf{x}), \quad i,j = 1,2 \,, \tag{8}$$

where $\alpha_L(\mathbf{x})$ is the longitudinal dispersivity in the principal flow direction. $\alpha_T(\mathbf{x})$ represents the transverse dispersivity, which is perpendicular to the longitudinal dispersivity. Notations $v_i$ and $v_j$ are the seepage velocities in different directions in the porous fractures and $\bar{v}$ represents the magnitude of the seepage velocity. In Eq. (8), $\boldsymbol{D}_0(\mathbf{x})$ is the effective molecular diffusion coefficient. In this study, $\boldsymbol{D}_0(\mathbf{x})$ in our model is assumed to be negligible, which implies that the dispersion is dominated by advective transport and mechanical dispersion (Zheng and Bennett, 2002).

## 2.2 DFN connections and the unstructured mesh generation

The information of fracture orientations enables the direct simulation of flow and transport in a series of 2D fractures and the reproduction of the flow and transport behaviors for a 3D DFN system. This study defines a DFN without isolated fractures as an effective discrete fracture network (DFNe). Figure 1 shows the definitions of the individual fracture in a 3D DFNe. Based on the long axis of an elliptical porous fracture, the positive trend and plunge angles are defined as clockwise from the north and downward from the horizontal plane, respectively. In this study, the intersections of a fracture and the simulation boundaries have to be identified (Figure 1) before the mesh generation is implemented. We generate the fracture length for the long and short axes of each fracture in the 3D DFN based on the uniform distribution. The larger value of the two generated radii is used to obtain the long axis of the elliptical fractures. In addition, isolated fractures that are not connected to other fractures and simulation boundaries are removed for computational efficiency.

Figure 2 shows an example of DFNe connection for two intersected fractures. Figure 2a shows the generated mesh in the 3D domain for two intersecting fractures. Mesh generation begins with the generation of initial fracture meshes for each fracture plate (i.e., Fractures 1 and 2 in Figures 2c and e, respectively). Figure 2b displays two intersecting fractures that were





individually rotated back to the 2D horizontal plane. In Figure 2b the plunge and trend values differ for each fracture in a 3D DFNe, so the fractures in a 2D coordinate system might not overlap. The intersections for each fracture are also located in different areas (see Figure 2b). However, the length of the intersection should be identical to that of the intersecting fractures. The fracture intersections and simulation boundaries are recorded for our unstructured mesh generation model.

The mesh generation starts with generating initial mesh for each fracture. The mesh generator allows users to define intervals of mesh boundary nodes. In this study, the Delaunay triangulation algorithm is used for generating the initial meshes. The special treatment of fracture intersections rely on the boundary recovery algorithm. In the process of boundary recovery for fracture intersections, we allow the node interval be reduced with a defined ratio according to the smallest node interval along the edges of the connected fractures. The ratio can be one-tenth or a smaller value depending on the problem. The

details of the mesh generation algorithm for connected fractures are available in the study of Lee and Ni (2015).

## 2.3 Numerical finite element solution to the DFNe

To solve the governing equations of flow and transport for the DFNe framework, we employ the Galerkin finite element method and the Biconjugate gradient matrix solver to solve Eqs. (1) and (4). The linear function for hydraulic heads and the concentration at the nodes surrounding an element of the triangular mesh system can be respectively represented with

$$h(\mathbf{x}) = \sum_{\alpha=1}^{3} h_\alpha N_\alpha(\mathbf{x}) ,$$ (9)

and

$$c(\mathbf{x}) = \sum_{\alpha=1}^{3} c_\alpha N_\alpha(\mathbf{x}) ,$$ (10)

where the notation $N_\alpha(\mathbf{x})$ is the shape function determined by the mesh in fractures in the 2D porous fracture plates in a 3D DFNe, and the shape function has the following formula:

$$N_\alpha(\mathbf{x}) = a_0 + \sum_{i=1}^{2} a_i x_i ,$$ (11)

Where $a_0$ and $a_i$ are coefficients of the shape functions. In this study, the coefficients were determined based on the following formulas as applied to nodes in a triangular element:

$$\{a_0\} = \frac{1}{J} \begin{Bmatrix} x_{12}x_{23} - x_{13}x_{22} \\ x_{13}x_{21} - x_{11}x_{23} \\ x_{11}x_{22} - x_{12}x_{21} \end{Bmatrix} ,$$ (12)

$$\{a_1\} = \frac{1}{J} \begin{Bmatrix} x_{22} - x_{23} \\ x_{23} - x_{21} \\ x_{21} - x_{22} \end{Bmatrix} ,$$ (13)





$$\{a_2\} = \frac{1}{J} \begin{Bmatrix} x_{13} - x_{12} \\ x_{11} - x_{13} \\ x_{12} - x_{11} \end{Bmatrix},$$  (14)

where $J = 2 \times A$, and $A$ represents the area of an element. Notation $x_{ij}$ represents the coordinate value in the $i$-direction at the $j$th node in a triangular element. In this study, the solution to the DFNe followed similar processes to what are used in classical finite element methods. However, the system of equations to be solved in the DFNe is different and relatively

complex compared to a typical 2D problem because the nodes along the fracture intersections in the solution processes could introduce additional terms in the coefficient matrices for the system of equations. The roles of the additional terms in the coefficient matrices are to build connections between fractures through the element nodes along the intersection lines. More intersections in the DFNe would yield more complex coefficient matrices for the system of equations. Figure 2c to Figure 2e also demonstrate an example for the connection of nodes and elements along a fracture intersection. Suppose that the

Fractures 1 and 2 in Figure 2c and Figure 2e have an intersection line (dashed line). Elements 1 to 5 in Fracture 1 and elements 6 to 10 in Fracture 2 share same nodes (nodes 48, 49, and 50) along the intersection line. Let us focus on the node 49. For all the elements in Fracture 1 and Fracture 2 that are connected to the node 49, the calculation of coefficient matrix for node 49 must rely on integrating weightings from shape functions of the connected elements and the associated nodes. This inclusion of all nodal information in the matrix system for heads could resolve the detailed behavior of flow in elements

that are connected to the fracture intersections. The mass flux of concentration near the intersections follow a similar procedure to build the coefficient matrices for the ADE solutions.

## 3 Transport model verification and numerical examples

The features of the HYDROGEOCHEM model are not for DFN flow and transport modeling. For simple cases such as

single horizontal fracture plate or cross-shaped porous fracture, it is possible to simulate fracture and matrix system using the HYDROGEOCHEM model if one can use small mesh sizes to resolve the fracture apertures and matrix system. In addition, the differences of the hydraulic conductivity between the fracture and matrix need to be large enough to minimize the influence of the matrix. Because the HYDROGEOCHEM was developed based on the finite element method, the numerical dispersion might be similar to the developed model in this study. This study further uses a two-dimensional analytical

solution proposed by Wexler (1992) to conduct verification of the developed model. The comparison is limited to the case with advection and dispersion in a horizontal porous fracture plate.

Based on the verified DFN flow and transport model, we then conducted 500 MCS realizations to assess the up-scaled flow behaviors with various fracture intensities for 2D profiles (i.e., $P_{21}$) and 3D rock volumes (i.e., $P_{32}$). The MCS flow realizations are further used to assess the effects of DFN properties on the flow and transport uncertainties. In this study,





statistical structures that are relevant to the distributions of the fracture properties included Poisson and uniform distributions (Lee and Ni, 2015; Xu and Dowd, 2010).

## 3.1 Transport model verification by using HYDROGEOCHEM model

This study employs two cases in a 2 m × 2 m × 2 m fractured rock domain (Figure 3), including a horizontal fracture plate (Case 1) and a cross-shaped fracture network (Case 2), to verify the developed transport model by using the HYDROGEOCHEM model. The test cases represent fracture sizes that are considerably larger than the controlled modeling domain. Local mesh refinements in the HYDROGEOCHEM model are required to resolve fractures in a rock volume to obtain fracture structures in HYDROGEOCHEM (see Figures 3a and c). However, the hydraulic conductivity values are different to represent a horizontal fracture plate (Figure 3a) and a cross-shaped fracture network (Figure 3c). For the two test cases, we assume a uniform fracture aperture of 0.001 m. The fracture hydraulic conductivity is 1.0 m/d for the fracture plates in the developed model. This fracture hydraulic conductivity value in HYDROGEOCHEM model is also applied to the elements that represent the fracture locations (Figures 3a and c). The matrix hydraulic conductivity in the HYDROGEOCEHM model is $10^{-5}$ m/d to create a high variety of hydraulic conductivity values for the fractures and rock matrix. In the test cases, the effective porosity for the fractures is at a relatively large constant value of 0.43. A relatively small isotropic dispersivity of 0.001 m is used to evaluate the advection dominated transport. The boundary conditions along the boundaries parallel to the flow direction are specified to be no-flow boundary conditions, except for the cross-shaped fracture network case, where a slightly upward flow along the vertical fracture is introduced (Figures 3c and d). The Neumann boundary condition is assigned in these cases as the transport boundary conditions. In the test cases, we release an initial Gaussian distribution plume in the horizontal fracture plate. The time step for the transport simulation is 0.1 days throughout the simulation time (5.0 days). Similar to the transport solution in the developed model, the ADE solution technique in the HYDROGEOCHEM model is the Eulerian-based approach for comparison purposes. Table 1 lists the flow and transport parameters for the test cases.

## 3.2 Transport model verification by using analytical solution

The study of Wexler (1992) considers a horizontal two-dimensional domain. The simulation domain is similar to the first case in section 3.1. However, the study of Wexler (1992) considers an x-direction uniform flow and constant values of longitudinal and transverse dispersion coefficients in the simulation domain. The transport process in the solution of Wexler (1992) also involves the first-order decay. The transport equation has the following formula:

$$\frac{\partial c(\mathbf{x},t)}{\partial t} = D_x \frac{\partial c(\mathbf{x},t)}{\partial x^2} + D_y \frac{\partial c(\mathbf{x},t)}{\partial y^2} - v_x \frac{\partial c(\mathbf{x},t)}{\partial x} - \lambda c(\mathbf{x},t) , \qquad (15)$$

with initial and boundary conditions:

$$c(\mathbf{x},0)\big|_\Omega = 0 , \qquad (16)$$





$$c(\mathbf{x},t)\big|_{x=0,\,Y_1 \le y \le Y_2} = c_D \,, \tag{17}$$

$$c(\mathbf{x},t)\big|_{x=0,\,y<Y_1,\,Y_2<y} = 0 \,, \tag{18}$$

$$\frac{\partial c(\mathbf{x},t)}{\partial x}\bigg|_{x=\infty} = 0 \,, \text{ and} \tag{19}$$

$$\frac{\partial c(\mathbf{x},t)}{\partial y}\bigg|_{y=\pm\infty} = 0 \,, \tag{20}$$

where the $c(\mathbf{x},t)$ is the concentration and $D_x$ and $D_y$ are the longitudinal and transverse dispersion coefficients. The $v_x$ is the uniform seepage velocity in x-direction. Notation $\lambda$ in Eq. (15) is the first-order decay coefficient for the model. In the test example, the specified concentration $c_D$ is applied along the inlet boundary (i.e., x=0) in the interval of $Y_1$ and $Y_2$. To fit the condition of the model in this study, we have neglected the decay process for comparison purpose. This approximation yields the closed form solution:

$$c(\mathbf{x},t) = \frac{c_D x}{\sqrt{\pi D_x}} exp\left(\frac{v_x x}{2 D_x}\right)$$
$$\cdot \int_{\tau=0}^{\tau=t} \tau^{-\frac{3}{2}} exp\left[-\left(\frac{v_x^2}{4D_x}\right)\tau - \frac{x^2}{4D_x \tau}\right] \cdot \left\{ erfc\left[\frac{Y_1-y}{2\sqrt{D_y \tau}}\right] - erfc\left[\frac{Y_2-y}{2\sqrt{D_y \tau}}\right]\right\} d\tau \tag{21}$$

The calculation of Eq.(21) requires numerical approximations. The study of Wexler (1992) suggested the Gauss-Legendre iteration algorithm to obtain the solution. However, their results indicated that a small x value might lead to numerical errors for the iterations.

In the test example (Case 3), the horizontal porous fracture plate has the size of 2m x 2m. We follow the assumptions applied to the solution in the study of Wexler (1992). The zero concentration is used to be the initial condition and the concentration of 1.0 is specified in the interval between $Y_1 = 0.75$ m and $Y_2 = 1.25$ m along the inlet boundary (i.e., x = 0). The uniform seepage velocity in x-direction is 0.1 m/d. The longitudinal and transverse dispersion coefficients are 0.005 m²/d.

### 3.3 Flow upscaling behaviors

The equivalent hydraulic conductivity for a specified representative elementary volume (REV) is the basis for conducting flow upscaling for practical problems that cover simulation domains on the order of hundreds of meters to several kilometers. Similar to the fractured rock volume in the test case, we generate 500 DFN realizations to assess the flow and upscaling behaviors for various fracture intensities and the associated fracture properties (e.g., fracture locations, plunges and trends, and sizes). Table 2 lists the parameters for generating the DFN realizations. In this study, the $P_{21}$ values for different DFN realizations are calculated based on the downstream boundary profile at x = 2.0 m (Figure 4). In the numerical example, the criteria of the nodal spaces along the fracture boundaries are fixed at 0.1 m, and the nodal spaces for fracture intersections have a reduction rate of one-tenth for all the DFNe realizations. Figure 4 also shows the conceptual model alongside one



DFNe realization and the associated head field as a numerical example. Table 3 lists the hydrogeological parameters for the flow and transport simulations.

The calculation of the equivalent hydraulic conductivity for a rock system considers the concept of mass conservation applied to a REV. The flow passing through the 3D control volume can be represented by the formula:

$$Q_{frac} = -K_{eq} \frac{\Delta h}{\Delta L} \cdot A \,, \tag{22}$$

where $Q_{frac}$ [L³/T] is the net flow rate for a specified direction of a DFNe in a defined rock control volume. The hydraulic head gradient of a control volume is defined as the ratio of the head difference $\Delta h$ [L] to the flow length $\Delta L$ [L] between two boundaries of the control volume. The notation $A$ [L²] represents the flow area for the control volume. The equivalent hydraulic conductivity $K_{eq}$ can be determined if the flow rate of a fractured rock is obtained from the DFNe flow model.

## 3.4 Sampling volumes and flow and transport uncertainties

In this numerical example, we investigate the effects of averaging strategies (i.e., along vertical lines or on profiles perpendicular to flows) on the observations of BTCs in fractured formations. This numerical example involves using the random DFNe flow realizations from the example in section 3.3. Table 3 lists the parameters applied to the transport simulations. Figure 5 shows the boundary conditions for the transport simulations. Constant concentration values of 1.0 and 0.0 (kg/m³) are specified on boundaries at x = 0.0 and 2.0 (m), respectively. The remaining boundaries are zero-concentration flux boundary conditions. Each time step in the simulation produces 500 concentration values at the computational nodes based on the 500 DFNe realizations. Then, we collect the concentration values and present the BTCs with means and standard deviations (STDs) to provide the variation bandwidths.

The x- and y-coordinates for the vertical sampling lines of concentration are (0.5, 1.0), (1.0, 1.0), and (1.5, 1.0). We allow the vertical lines to have a fixed diameter to represent wells in practical problems and the wells are assumed to be opened throughout the rock volume (Figure 5). The BTCs along the wells are calculated by using flux weighted concentration (i.e., $v(\mathbf{x},t)c(\mathbf{x},t)$ ) at nodes involved in the wells. In this study, the well diameter is fixed to 4 in. to calculate the mean BTCs and BTC uncertainties along the wells.

We further define four profiles to assess the flow and transport uncertainties. The profiles can be considered as a series of wells installed along the profiles. The vertical profile (y = 1.0 m) along the flow direction (see Figure 5) is used for assessing spatial variability in flow and transport. This y = 1.0 m profile has the width of 4 in. along the profile. To obtain the head and transport uncertainties along the vertical profile at each x location, we average the solutions of flow and transport along the z-direction. The sampling x step is the same as the profile width. Solutions of MCS at each x location are collected based on the nodal heads and concentration fluxes in the cuboid.

Three profiles perpendicular to the flow direction have the locations x same as the sampling wells, i.e., x = 0.5 , 1.0, and 1.5 (m). In this study, we also consider the profiles to have the widths same as the well diameter (4 in.) for comparing vertical



line (i.e., the well) and profile (i.e., a series of wells) sampling strategies. The representative heads and BTCs of the profiles are calculated by averaging the fracture nodal heads and concentrations in the profile volumes.

## 4 Results and Discussion

This study focused on a relatively small fractured rock volume that was 2 m × 2 m × 2 m in size for the model verification and stochastic flow and transport modeling. The modeling domain might not be limited to this size when implementing this model. The following sections provide the results of the transport model verification and the uncertainties obtained from realizations of MCS.

### 4.1 Transport model verification

Figures 6 and 7 show a comparison of the concentration distributions for the horizontal fracture plate and cross-shaped fractures. The results in Figures 6 and 7 show that identical solutions were obtained from both the developed DFN model and the HYDROGEOCHEM model. All the temporal and spatial variations in the plume were determined, and the solutions from the developed and HYDROGEOCHEM models were found to be identical. Figure 7 shows the concentration distributions after 3.0 days (Figures 7a and b) and 5.0 days (Figures 7c and d) when using the HYDROGEOCHEM and developed models for a cross-shaped fracture system. With a specified small upward flow applied to the vertical fracture, portions of the concentrations moved upward near the fracture intersection line (Figures 7b and d). This slightly upward flow relied on the constant head of 9.01 m that was applied on the top side of the vertical fracture. Again, the developed and HYDROGEOCHEM models were found to be identical for the cross-shaped fracture network.

Figure 8 shows the comparison of solute concentration obtained from analytical (dashed lines) and the developed (solid lines) models. Figures 8(a) to (d) show the concentration distribution at time 5.0, 7.5, 10.0, and 15.0 days, respectively. The concentration for the contour lines are for the relative concentration ($c/c_D$) of 0.1, 0.5, and 0.7. The continuous release of concentration along the central portion of the x = 0 boundary leads to a high concentration area near the release location. In the test example the uniform flow was assigned for the fracture plate. The longitudinal dispersion is relatively obvious as compared to the transverse dispersion. In summary, the results of the developed model agree with those from the analytical model.

### 4.2 Flow upscaling behaviors

Figure 9 shows the results of the flow simulations when applied to the 500 DFN realizations. Figure 9a shows that the number of fractures increased with the effective fracture intensity $P_{32}^e$. A higher number of fractures could create greater variations in the 3D fracture intensity. However, such behavior was not valid for the 2D fracture intensity $P_{21}^e$ (see Figure 9b).



Similar numbers of fractures might vary in a wide range of $P_{21}^e$ values. This is because of the $P_{21}^e$ calculation that relies on counting the trace length (i.e., fracture intersections between fractures and the rock volume boundary) on the downstream boundary profile (i.e., x = 2.0 m) of the simulation domain. We found that different sampling profiles at fixed x locations exhibited similar patterns for the comparison of variables $P_{21}^e$ and $P_{32}^e$. Figure 9c summarizes the variations in the equivalent

hydraulic conductivity with different fracture intensities. The results revealed that the high fracture intensity generally created a high equivalent hydraulic conductivity. The results in Figure 9c shows that the equivalent hydraulic conductivity values were 2 to 3 orders of magnitude lower than the specified fracture hydraulic conductivity value. Figure 9d further shows the comparison of $P_{32}^e$ and $P_{32}^t$. The results showed that small numbers of fractures can result in large variations of $P_{32}^t$ as compared to a known $P_{32}^e$. Such results implied that the fracture diameter used in the DFN generations might be relatively

small so that many isolated fractures were removed for flow and transport simulations. The behavior could also led to high variations of flow and transport simulations.

### 4.3 Sampling volumes and flow and transport uncertainties

Figure 10 shows a realization of the transport simulation based on the DFNe flow field in Figure 4. These results revealed that the continuous concentration released on the left boundary gradually migrated along the connected fractures. The spatial

distributions of the concentration on the fracture plates were highly variable. This finding validated the concept that was proposed by Park et al. (2003), who stated that local flow cells contribute less to flow and contaminant transport in fracture formations.

Figure 11 shows the mean concentration BTCs (solid lines) and the associated STD intervals (i.e., ± STDs from the means) at sampling wells (Figures 11a to 11c) and along profiles (Figures 11d to 11f) at location x = 0.5 m, 1.0 m, and 1.5 m,

respectively. Comparisons of the means and ± STDs BTCs at the wells and along the profiles are shown in Figures 11g to 11i, where the solid lines represent the mean concentrations along the wells and the dashed lines are the average concentrations along the profiles. We collected 500 BTC realizations (gray lines) from the flow simulations and recursively estimated the means and STDs of the concentrations at different time steps. The results in Figure 11 clearly show that the maximum mean concentration BTCs for the different averaging strategies might not reach the maximal concentration that is

released on the boundaries. The maximum concentrations of the mean BTCs were 60% to 80% of the concentration at the maximum released source. A longer simulation time might be required to obtain the maximum concentration. Similar results were obtained with the same averaging strategy at different monitoring locations. In general, small sampling volume (i.e., the wells) obtain relatively large values of concentration means for specified times. In addition, such small sampling volume can also lead to higher variations of BTCs (i.e., STDs). In this study, the differences between two sampling strategies were not

significant.

Figure 12 shows the distributions flow uncertainties along the flow direction. Please note that the results were based on collections of vertical averaged heads and Darcy velocities at each x location along y = 1.0m. The results in Figure 12



showed high nonstationarity of the head and x-direction Darcy velocity. The distribution of the head variance exhibited high variation at the inlet boundary and the head variance gradually decreased to a small value at the outlet boundary (Figures 12a and b). This was an interesting result because the behavior was different from cases in heterogeneous porous media, which showed similar head variations near inlet and outlet boundaries (Li et al., 2004; Ni and Li, 2005, 2006). We believed that the

extremely high head variations near DFN inlet boundary could be induced by the generated DFN realizations. In the DFN flow simulation, the constant head on the inlet boundary were applied to the fractures that connect to the inlet boundary. The number of intersections at boundary controlled the inlet flow for a generated DFN. Therefore, the uncertainty of head must involve the uncertainty of fractures connected to the inlet boundary.

Figures 12c and d show the distribution of velocity variance in x-direction. In general, the highest value of the x-direction

velocity variance was one order of magnitude smaller than the highest value of the head variance. The x-direction Darcy velocity also showed high variations near inlet and outlet boundaries. However, we found that the velocity variance along the x-direction showed a relatively stationary zone away from the inlet and outlet boundaries. This behavior had been observed in groundwater flow in porous media (Ni et al., 2010, 2011). The distance of the variance transition zone was close to 0.5 m from the boundaries. Such value was close to the mean fracture diameter used for generating DFN realizations. Figures 12e

to 12h show the means and variances of the Darcy velocities in the y- and z-directions. The results indicated that the velocity variations in the y- and z-directions were relatively stationary. We found that the boundary-induced nonstationarity were not obvious for velocity variations in y- and z-directions. The stationary variances of the velocities in y- and z-directions were similar to that of the x-direction velocity. In Figure 12 the fluctuations of head and velocity variances indicated that more DFN realizations might be required to obtain smooth distributions of head and velocity variances.

Figure 13 shows the distributions of transport uncertainty at different times along the centerline profile (y = 1.0m). This study specified a constant concentration on the inlet boundary. Similar to the situation of the constant head condition in flow simulations, transport results also showed high nonstationarity near the inlet boundary. However, the variances can propagate to the downstream area with time. In the transport simulations, we specified a constant concentration of 0 (kg/m$^3$) on the outlet boundary. The uncertainty of concentration at the outlet boundary increased with times because of the arrivals

of concentration fronts. Similar to the condition on the inlet boundary, the number of fracture intersections might influence the increase of the concentration variation at the outlet boundary.

## 5 Conclusions

This study developed finite element flow and advection and dispersion transport models for complex 3D DFNs in fractured

formations. When testing transport model, identical temporal and spatial solutions were obtained from the developed model and the HYDROGEOCHEM model based on a Gaussian-type initial plume that was released in the porous fracture plate. For a simplified case that exists an analytical 2D transport solution. The developed model produced accurately the results of



concentration distributions in a horizontal fracture plate. The MCS flow simulations showed that different fracture intensities can result in variations in the equivalent hydraulic conductivity that were 2 to 3 orders of magnitude lower than the fracture hydraulic conductivity values.

Simulations of transport in 3D DFNs revealed that the maximum concentration of mean BTCs for different averaging strategies might not have reached the concentration. The sampling strategies along the wells and profiles yielded similar BTCs patterns. Based on the MCS, the means and STDs for the two sampling strategies were observed at different sampling locations. MCS results showed that a smaller sampling volume can lead to relatively large values of mean concentrations and concentration STDs for specified times.

MCS flow and transport showed that the distribution of the head variance exhibited high variations at the inlet boundary and the head variance gradually decreased to a small value at the outlet boundary. The extremely high head variation near DFN inlet boundary could be induced by the generated DFN realizations. No stationary zone for the head variance was obtained based on collected MCS realizations. In the study, the value of the highest x-direction velocity variance was one order of magnitude smaller than the value of the highest head variance. The velocity variance along the x-direction showed a relatively stationary zone away from the inlet and outlet boundaries. The distance of the variance transition zone was close to 0.5 m, which was the value of mean fracture diameter for DFN generations. Results of the velocity variations in the y- and z-directions showed relatively stationary along the flow direction. The boundary-induced nonstationarity were not obvious for velocity variations in y- and z-directions. Results of flow uncertainties showed that more DFN realizations might be required to obtain smooth distributions of head and velocity variances. The MCS of transport modeling showed high nonstationarity of concentration variation near the inlet boundary and the variance can propagate to the downstream with time.

**Acknowledgments**

This research was partially supported by the Ministry of Science and Technology, Republic of China under grants MOST 103-2221-E-008 -049 -MY3, NSC 102-2116-M-008 -010, and NSC 100-2625-M-008 -005 -MY3, and by the Institute of Nuclear Energy Research, Atomic Energy Council, Executive Yuan under grant NL1030099, and by the Soil and Groundwater Pollution Remediation Fund in 2017.

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



**Table 1: The flow and transport parameters that were used for the transport verification cases.**

| | Case 1 & 2 | Case 3 |
|---|---|---|
| West B.C. | $h = 9.1\,\text{m} / \nabla c = 0$ [a] | $c_D = 1.0$ kg/m$^3$ ( $0.75 \le y \le 1.25$ ) <br> $c_D = 0$ kg/m$^3$ ( $y < Y_1$, $y > Y_2$ ) |
| East B.C. | $h = 9.0$ m$/ \nabla c = 0$ | $\nabla c = 0$ |
| North B.C. | $\nabla h = 0 / \nabla c = 0$ | $\nabla c = 0$ |
| South B.C. | $\nabla h = 0 / \nabla c = 0$ | $\nabla c = 0$ |
| Top B.C. | $\nabla h = 0 / \nabla c = 0$ <br> $h = 9.01$ m (Case 2) [b] | N/A |
| Bottom B.C. | $\nabla h = 0 / \nabla c = 0$ | N/A |
| Fracture aperture (m) | 0.001 | N/A |
| K in fractures (m/day) | 1.0 <br> 2.0 | N/A |
| K in matrix (m/day) | 0.001 | N/A |
| Effective porosity(-) | 0.43 | N/A |
| Seepage velocity (m/day) | Variable [c] | $v_x = 0.1$, $v_y = 0.0$ |
| Isotropic dispersivity(m) | 0.001 | 0.05 |
| Time step (day) | 0.1 | 0.1 |
| Simulation time (day) | 5.0 | 15.0 |

[a] the specified boundary conditions for HYDROGEOCHEM and the developed model are applied to the intersection between fractures and the West (or East) boundary of the simulation domain.

5  [b] the specified boundary conditions for HYDROGEOCHEM and the developed model are applied to the intersection between the vertical fracture and the Top boundary of the simulation domain.

[c] the seepage velocity at each node is evaluated based on the Darcy flux obtained at the node.



**Table 2: Parameters that were used to generate 3D DFNs for the flow example.**

| Parameters | Value |
| --- | --- |
| $X_{min}/X_{MAX}$ | 0.0/2.0 m |
| $Y_{min}/Y_{MAX}$ | 0.0/2.0 m |
| $Z_{min}/Z_{MAX}$ | 0.0/2.0 m |
| Mean fracture intensity | 6.0 m$^{-1}$ |
| Trend (min/max) | 96.0/140.0 |
| Plunge (min/max) | 16.0/90.0 |
| Mean radius of ellipse | 0.5 m |

**Table 3: Parameters that were used for the transport simulations in the 3D DFNe realizations.**

| Parameters | Value |
| --- | --- |
| Fracture aperture | 0.001 m |
| Fracture intensity | 3 to 7 m$^{-1}$ |
| Fracture hydraulic conductivity | 0.43 m/day |
| Effective porosity | 0.43 |
| Isotropic dispersivity | 0.1 m |
| Simulation time step | 0.1 days |
| Total simulation time | 50 days |





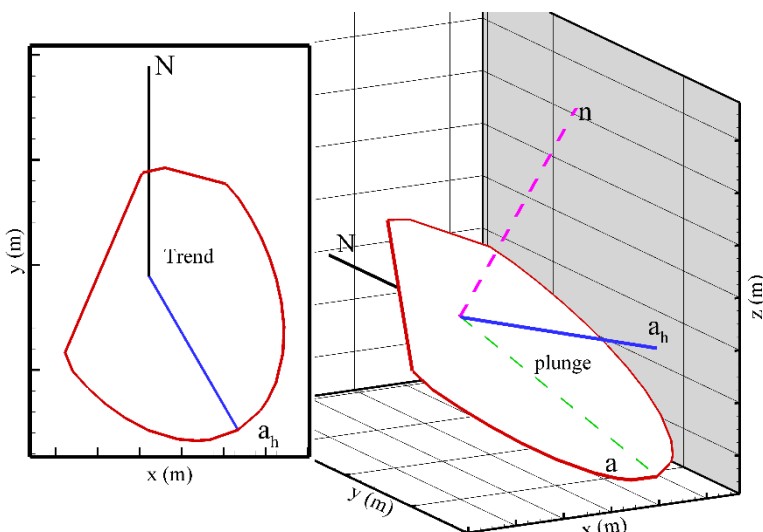

**Figure 1: Definitions of fracture orientation for a fracture in a 3D DFNe. In this study, the positive trend and plunge angles were clockwise from the north and downward from the horizontal plane, respectively.**





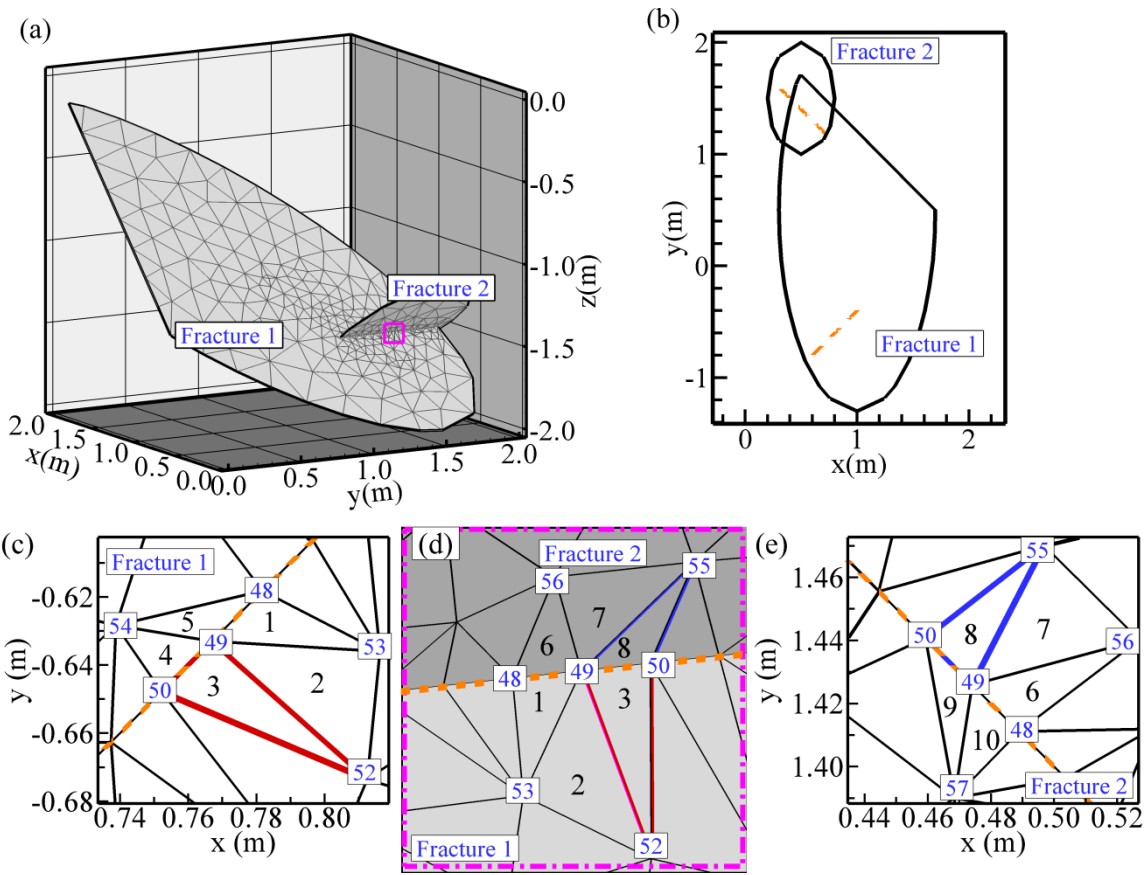

**Figure 2: Example of a generated mesh for two intersected fractures in (a) a 3D DFNe, (b) 2D plane view of back rotated fractures based on the individual fracture orientation, (c) the plane view of the mesh for Fracture 1 near the intersection, (d) the close view of the fracture intersection in a 3D DFNe, and (e) the plane view of the mesh for Fracture 2 near the intersection.**





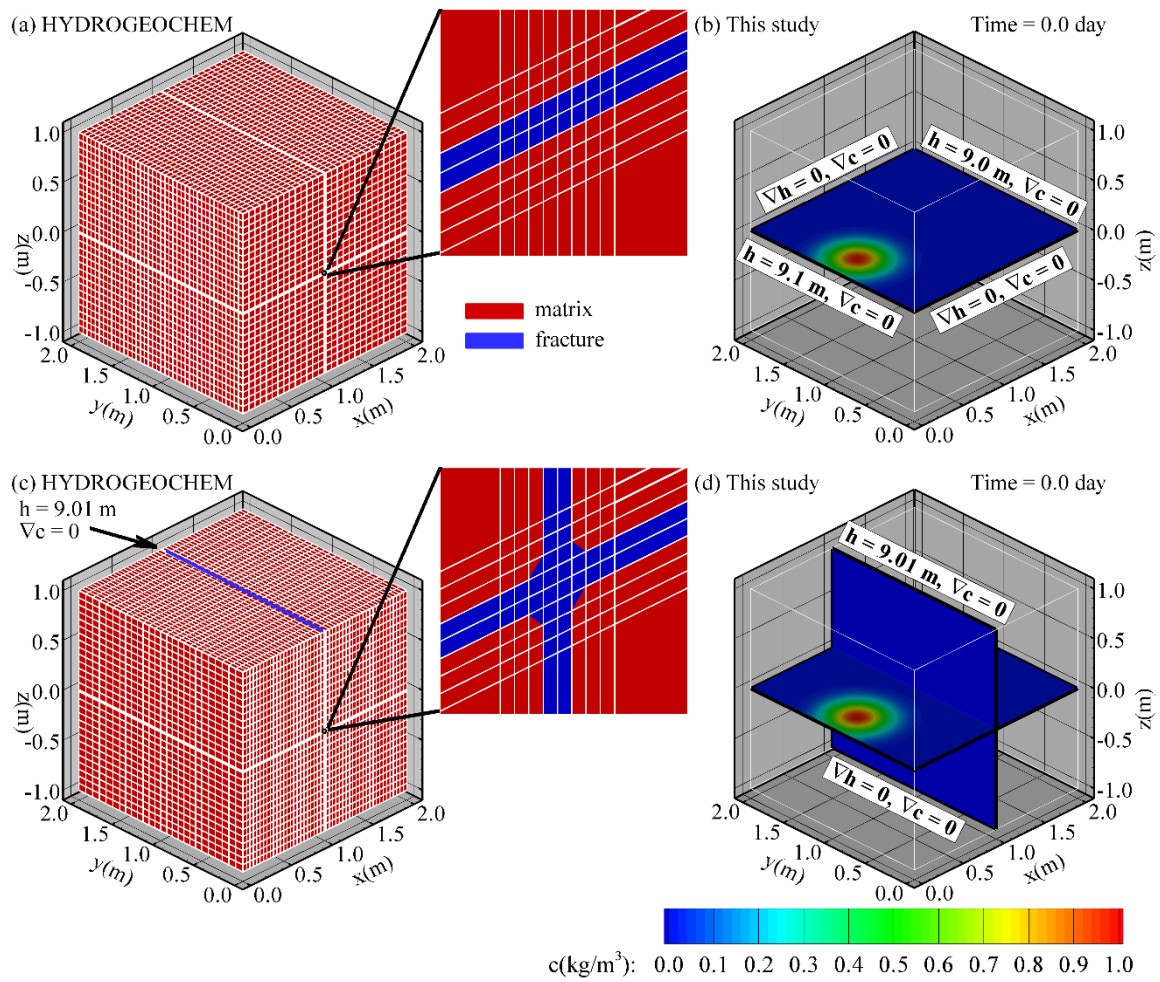

**Figure 3: Conceptual model for the verification of a 2D horizontal fracture plate and a cross-shaped fracture network: (a) the mesh of HYDROGEOCHEM model for the horizontal fracture plate, (b) the DFN conceptual model for the horizontal porous fracture, (c) the mesh of HYDROGEOCHEM model for the crossed-shaped fracture network, and (d) he DFN conceptual model for the crossed-shaped fracture network. Note that the single fracture plate (i.e., (a)) and cross-shaped fracture network (i.e., (c)) in the HYDROGEOCHEM model are represented with the relatively high hydraulic conductivity (i.e., 1.0m/d). However, the hydraulic conductivity for the matrix in HYDROGEOCHEM model was assumed to be 10⁻⁵m/d.**





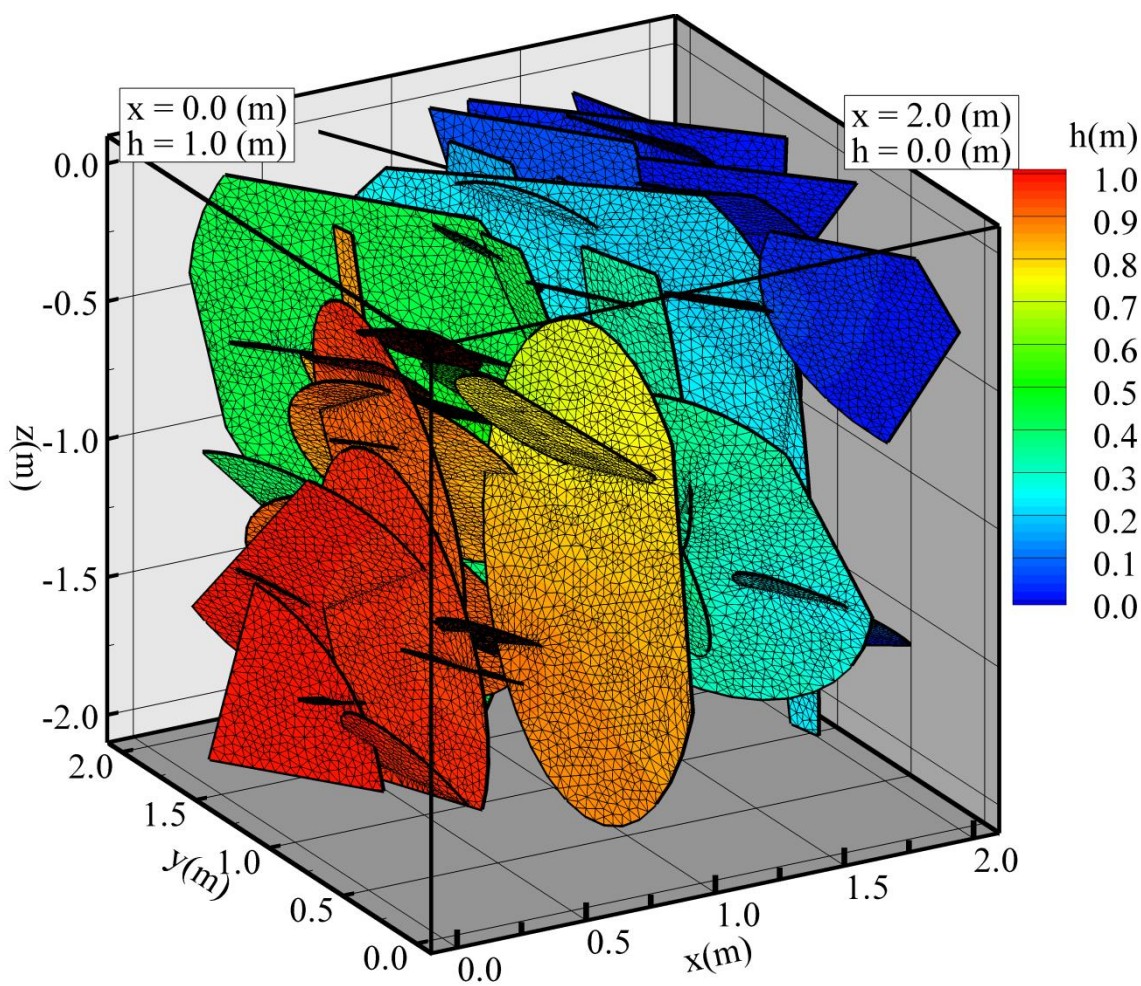

**Figure 4: Conceptual model, one DFNe realization, and the associated flow field for the numerical example.**




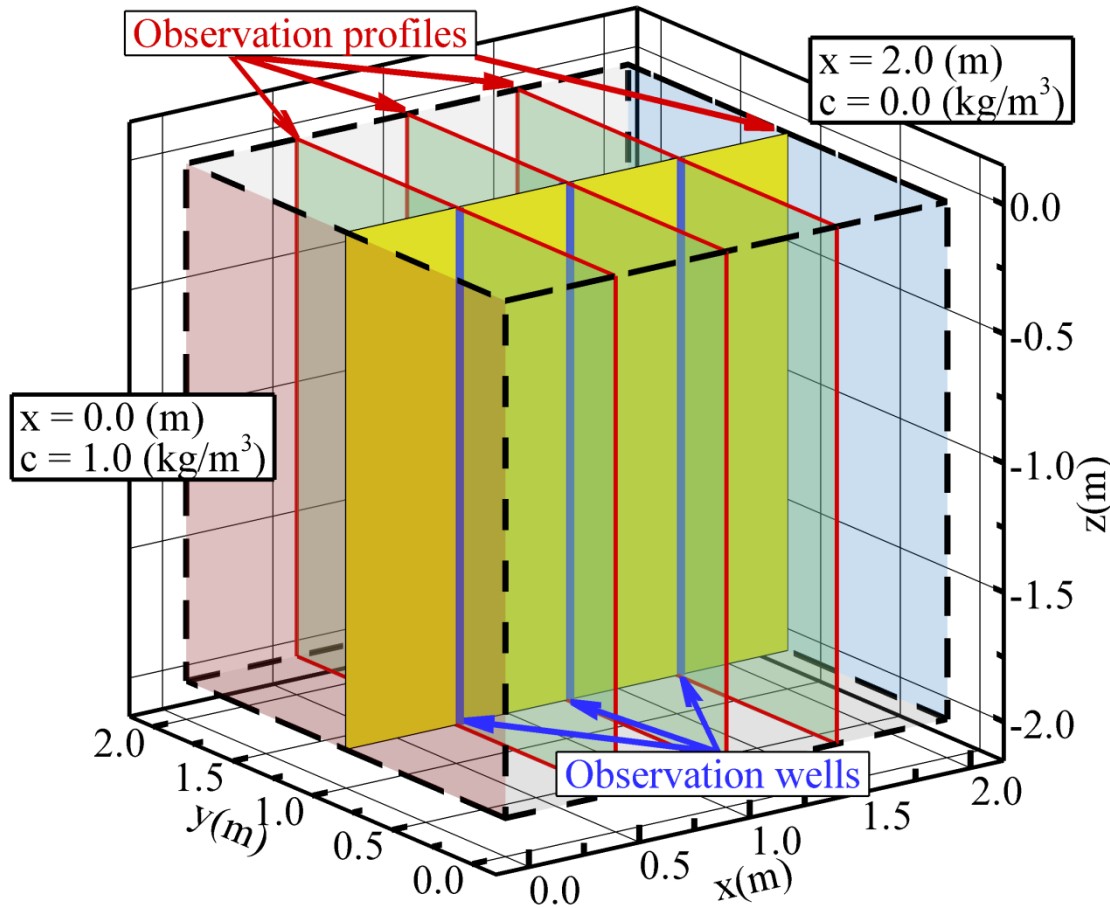

**Figure 5: Conceptual model and the specified well and profile locations for the calculations of the flow and transport uncertainties.**





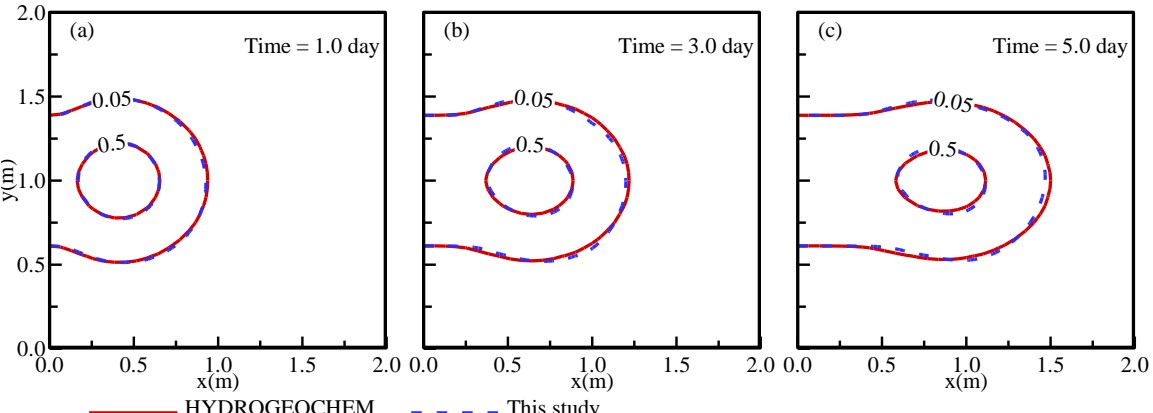

**Figure 6: Comparison of the concentration distributions for the developed DFN model (dashed lines) and the HYDROGEOCHEM model (solid lines) for a horizontal fracture plate.**




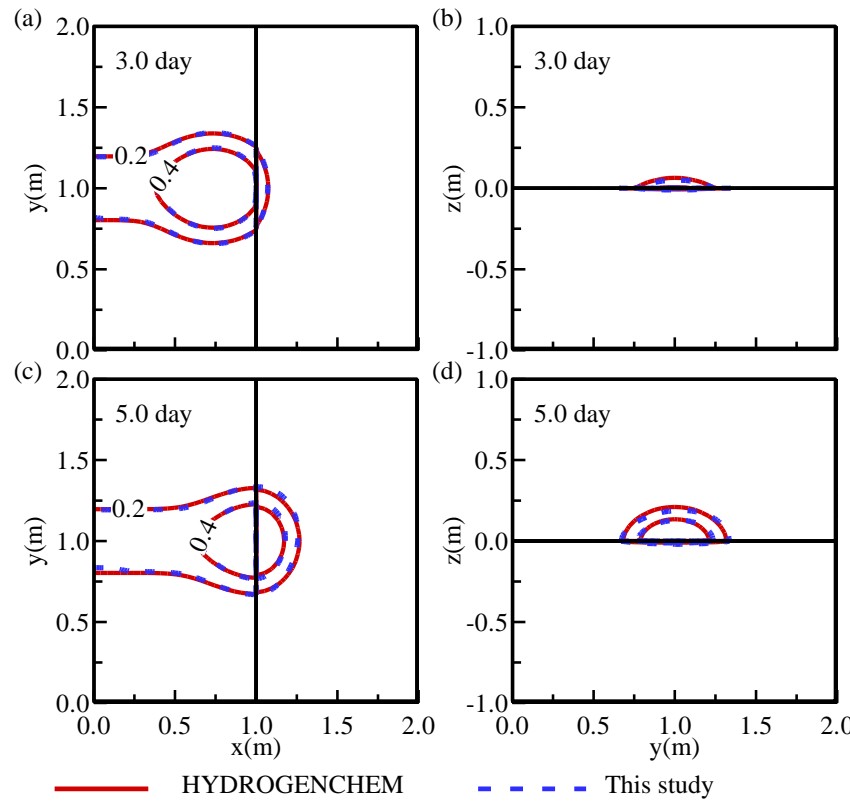

**Figure 7: Comparison of the concentration distributions for the developed DFN model and the HYDROGEOCHEM model for a cross-shaped fracture network: (a) top view of the horizontal fracture at t = 3.0 day, (b) front view of the vertical fracture at t = 3.0 day, (c) top view of the horizontal fracture at t = 5.0 day, and (d) front view of the vertical fracture at t = 5.0 day.**





**Figure 8: The comparison of solute transport for an inlet continuous source in a 2D horizontal porous fracture at time: (a) 5.0 days, (b) 7.5 days, (c) 10.0 days, and (d) 15 days.**



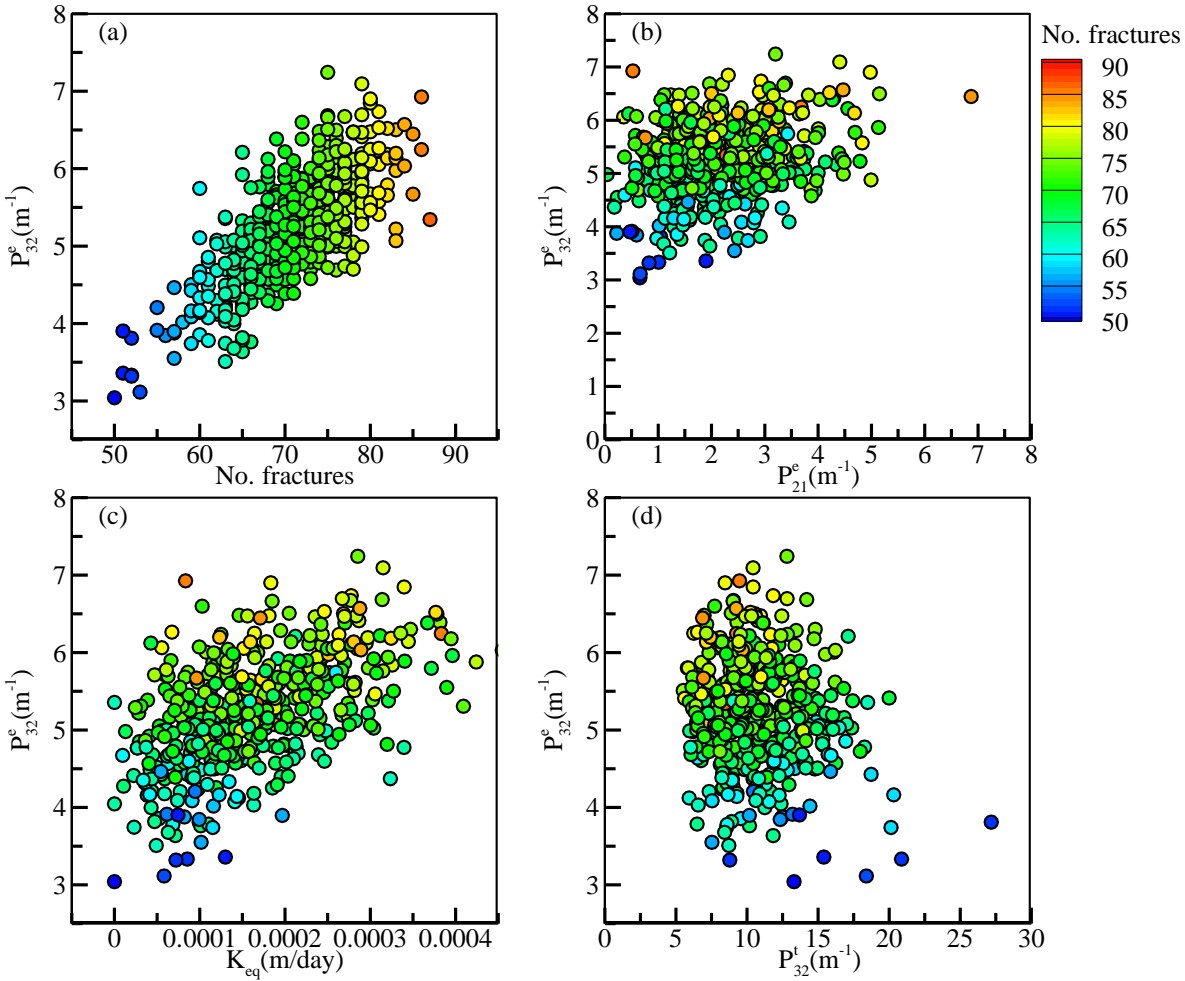

**Figure 9: Comparisons of the DFN properties and equivalent hydraulic conductivity for the 500 generated DFN realizations. The superscripts e and t represent the effective and total fracture intensities.**


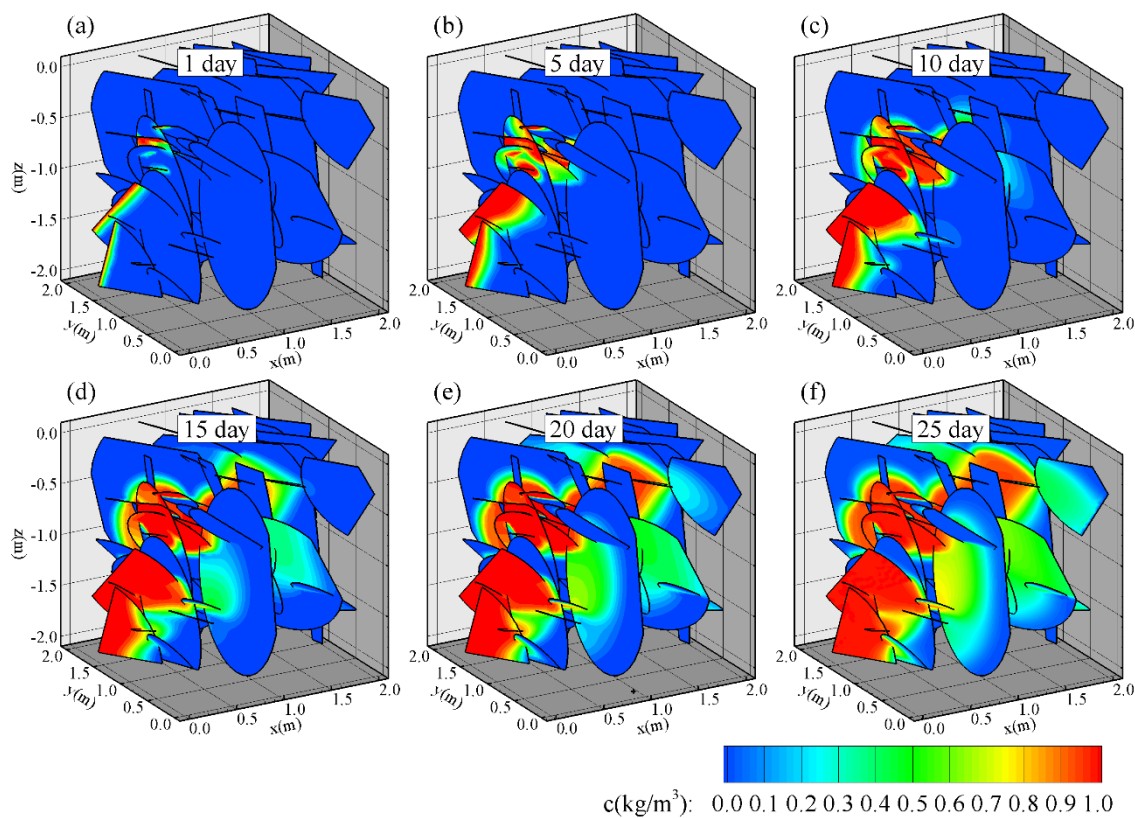

**Figure 10: A realization of the transport simulation based on the DFNe flow field in Figure 4.**





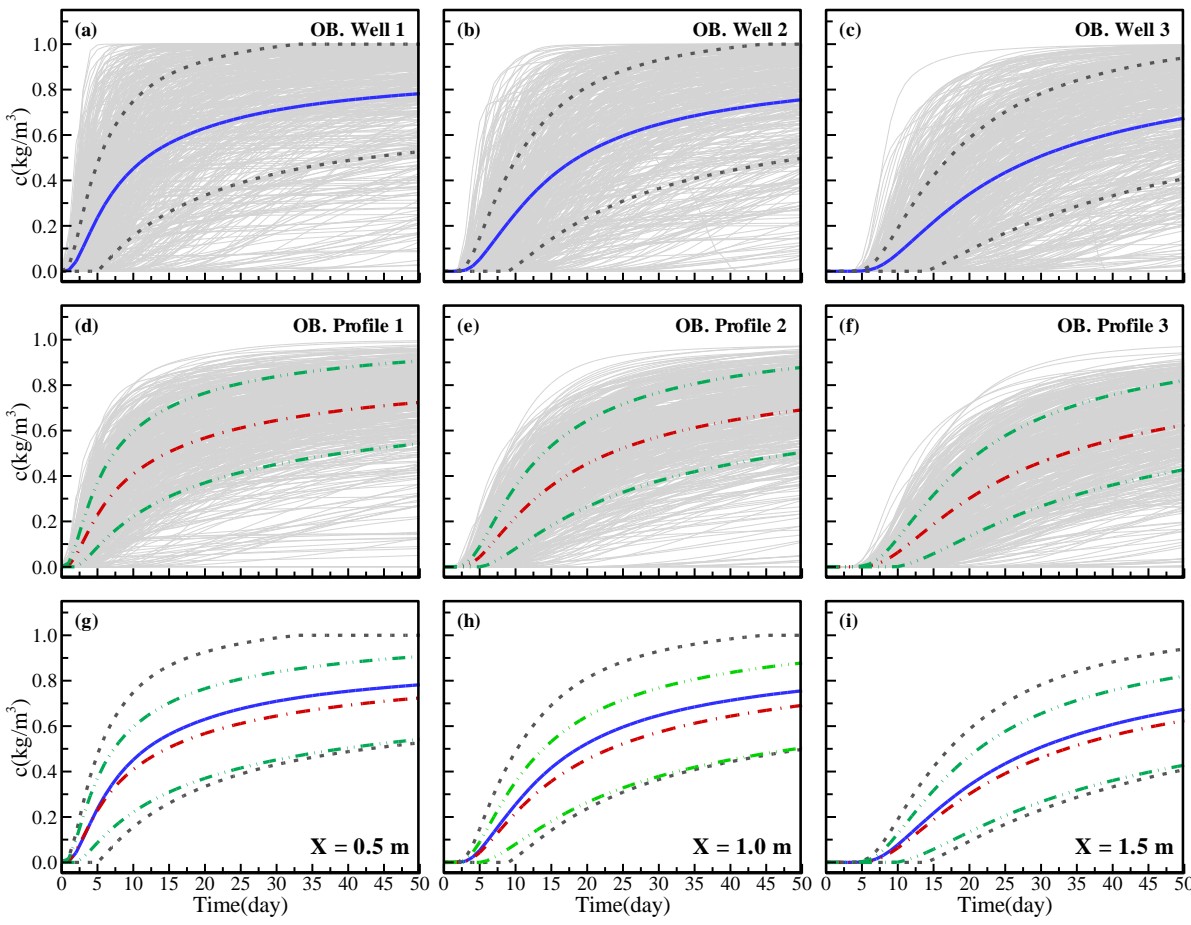

**Figure 11: Mean concentration BTCs (solid lines) and associated standard deviation intervals (dashed lines) at the sampling wells for (a) x = 0.5 m, (b) x = 1.0 m, and (c) x = 1.5 m and the mean BTCs (dash dotted lines) and STDs for the sampling profiles at (d) x = 0.5 m, (e) x = 1.0 m, and (f) x = 1.5 m. The BTC statistics, which were based on different averaging strategies, are presented at (g) x = 0.5 m, (h) x = 1.0 m, and (i) x = 1.5 m, where the solid lines in (g), (h), and (i) indicate the average BTCs along the wells and the dash dotted lines represent the average BTCs along the profiles.**





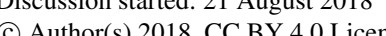

**Figure 12: Comparison of flow uncertainties for vertical profile along the flow direction: (a), (c), (e), and (g) the distributions of mean head and Darcy velocity and the associated STDs about the means; (b), (d), (f), and (h) the distributions of the head and velocity variances along flow direction. Note that the results were based on the vertical averaged head and Darcy velocities at each x location along the y = 1.0m profile. For each x location, the sampling volume was same as the size of the specified sampling well.**



**Figure 13: Transport uncertainties for the vertical profile along the flow direction: (a), (c), (e), and (g) the distributions of mean concentration and the associated STDs about the means at different times; (b), (d), (f), and (h) the distributions of the concentration variances along flow direction at different times. Note that the results were based on the vertical averaged concentration at each x location along y = 1.0m line.**