# Peer review of "Stochastic modeling of flow and conservative transport in threedimensional discrete fracture networks"

_Hydrology and Earth System Sciences, 2018_

## Referee Comment (RC1) · Anonymous Referee #1 · 19 Sep 2018

1) 3.1 Transport model verification by using HYDROGEOCHEM model 'The boundary conditions along the boundaries parallel to the flow direction are specified to be no-flow boundary conditions, except for the cross-shaped fracture network case, where a slightly upward flow along the vertical fracture is introduced' Please explain better the upward flow. Is it a constant head value of 9 m as shown by the figure?

2) 3.2 Transport model verification by using analytical solution On the basis of which criteria did you choose the dispersivities in the HYDROGEOCHEM model and the analytical solution?

3) 4.1 Transport model verification 'The longitudinal dispersion is relatively obvious as

compared to the transverse dispersion'. Explain this sentence.

4) Figure 9 Please define in a careful way the parameters P21 and P32 as well as Pe 21 Pe32 Pt32 as the notations are quite misleading. How did you calculate those parameters and do they differentiate among each other? And also better interpret the graph on the basis of those parameters and others such as fracture hydraulic conductivity and equivalent hydraulic conductivity. The discussion of graph 9d is not clear, please provide a more accurate interpretation of the results. 5) 5 Conclusions The Conclusion is a mere summary of the obtained results. Rewrite the conclusion adding a more extensive interpretation and discussion of the results, including clarifications on the novelty of the proposed approach and how it would provide a benefit to the scientific community.

Please also note the supplement to this comment:
https://www.hydrol-earth-syst-sci-discuss.net/hess-2018-397/hess-2018-397-RC1-supplement.pdf

---

## Referee Comment (RC2) · Anonymous Referee #2 · 1 Oct 2018

This is a very interesting topic, the efforts of the authors should be applauded by the community. The only concern i have is the practical application of this innovation. I am wondering if the authors can find experimental data to compare with their numerical calculations?

---

## Author Comment (AC2) · 7 Nov 2018

November 6, 2018

We thank Reviewer #1 for the valuable comments to our manuscript. The following is the list of the responses to the comments proposed by Reviewer #1.

1)
3.1 Transport model verification by using HYDROGEOCHEM model
'The boundary conditions along the boundaries parallel to the flow direction are specified to be no-flow boundary conditions, except for the cross-shaped fracture network case, where a slightly upward flow along the vertical fracture is introduced'
Please explain better the upward flow. Is it a constant head value of 9 m as shown by the figure?

**Response**:
Yes, the upward flow was created by a constant head assigned on the top boundary of the vertical fracture.
We will add detailed description for the B.C. of the case with cross-shaped fracture network. Additionally, the Figures 3c and d will be modified to make the concept clear.
Thanks for the suggestion.

2)
3.2 Transport model verification by using analytical solution
On the basis of which criteria did you choose the dispersivities in the HYDROGEOCHEM model and the analytical solution?

**Response**:
We proposed that the fractures have rough surfaces with numerous contact points and fractures consist of the void space enclosed between two impermeable surfaces, which in a topological sense constitutes a two-dimensional porous medium. Therefore, the concept of porous fracture plates was employed in this study to formulate three-dimensional DFNs (Pruess and Tsang, 1990). Such porous fracture plates enable the use of HYDROGEOCHEM model and the analytical model for validations
The isotropic dispersivity used for the test cases was 0.1m. This value was determined based on the study of scale-dependent dispersivity proposed by

Gelhar et. al.(1992). In our study, the interested scale is approximately on the order of 1m. Therefore, in our validation cases with the domain sizes of 2m, we use 0.1m to be the isotropic dispersivity for the transport simulation. Thank you for the comment. We will add the detailed information in the revised manuscript.

**References:**

Pruess, K., Tsang, Y.W., 1990. On two-phase relative permeability and capillary pressure of rough-walled rock fractures. Water Resour. Res., 26, 1915–1926.

Gelhar, L.W., Wetly, C., Rehfeldt, K.R., 1992. A critical review of data on field-scale dispersion in aquifers. Water Resour. Res., 28(7), 1955-1974.

3)

4.1 Transport model verification

'The longitudinal dispersion is relatively obvious as compared to the transverse dispersion'.   Explain this sentence.

**Response**:

Thank you for the comment. We will modify the discussion in the manuscript. In the comparison cases, the isotropic dispersivity was assigned for the simulation. Based on the x-direction uniform flow applied to the model, the effect of directional dispersion on the transport should be the same. The dispersion coefficient for x- and y-directions are:

$$\boldsymbol{D}_{xx}(\mathbf{x}) = \alpha_L(\mathbf{x})\frac{v_x v_x}{\overline{v}} + \boldsymbol{D}_o(\mathbf{x}),$$

(1)

and

$$\boldsymbol{D}_{yy}(\mathbf{x}) = \alpha_T(\mathbf{x})\frac{v_x v_x}{\overline{v}} + \boldsymbol{D}_o(\mathbf{x}),$$

(2)

where $a_L(\mathbf{x})$ is the longitudinal dispersivity in the principal flow direction. $a_T(\mathbf{x})$ represents the transverse dispersivity, which is perpendicular to the longitudinal dispersivity. Notations $v_x = v_x(\mathbf{x})$ and $v_y = v_y(\mathbf{x})$ are the seepage velocities in different directions in the porous fractures and $\overline{v}$ represents the magnitude of the seepage velocity. Notation $D_o(\mathbf{x})$ is the effective molecular diffusion coefficient.

In the test example, the contribution from the advective transport is mainly from the seepage velocity in the x-direction. We will modify the discussion

and revise the equations to make the presentation clear.

4)

Figure 9

Please define in a careful way the parameters $P_{21}$ and $P_{32}$ as well as $P^e_{21}$ $P^e_{32}$ $P^t_{32}$ as the notations are quite misleading.

How did you calculate those parameters and do they differentiate among each other? And also better interpret the graph on the basis of those parameters and others such as fracture hydraulic conductivity and equivalent hydraulic conductivity. The discussion of graph 9d is not clear. Please provide a more accurate interpretation of the results.

**Response**:

Thanks for the comments. The definitions for different fracture intensities will be defined clearly in the revised manuscript. Additionally, the calculations of these values will be presented with details.

In this study the parameters $P_{21}$ and $P_{32}$ are the length of fracture traces per unit area (2D domain) and the area of fractures per unit volume (3D domain), respectively. We add a superscription "e" to indicate the intensity of effective fractures. The effective fractures are those that neglect isolated fractures in a rock. The superscription "t" represents the total fracture intensity for a rock. The concept of the equivalent hydraulic conductivity is to support the discussion of up-scaling issue. The detailed information and discussion will be provided in the revised manuscript.

5)

5 Conclusions

The Conclusion is a mere summary of the obtained results. Rewrite the conclusion adding a more extensive interpretation and discussion of the results, including clarifications on the novelty of the proposed approach and how it would provide a benefit to the scientific community.

**Response**:

Thanks for the comments. Yes, there are novelty of the proposed approach and benefits to the scientific community.

Numerical solutions to the ADE based on the Eulerian approach have not

been widely implemented to DFN. This is because of computational issues such as numerical dispersion and convergence in the model for complex fracture connections. With the developed model, we also found interesting results that can contribute to the research communities of flow and transport in fractured rocks. We will follow the suggestions to rewire the conclusion.

---

## Author Response (AR1)

December 05, 2018

Dear Editor,

    We are pleased to submit our revised manuscript entitled "Stochastic modeling of flow and conservative transport in three-dimensional discrete fracture networks " for possible publication in Hydrology and Earth System Sciences. In the newly uploaded manuscript, we have revised the manuscript (marked with red color) based on the review comments from Editor and two reviewers. The review comments are valuable to improve the presentation of the paper. The following is the list of point-by-point responses to the comments and suggestions.

**Editor comments and suggestions:**
1)
- How did you define the number of MC simulations (500), can you provide some quantitative information that helped you to define this number?

**Response**:
The results of MC rely on the number of DFN realizations. With a well-developed model, more MC realizations can lead to a more accurate result. In this study, we focused on assessing flow uncertainties induced by the specified DFN structures. The acceptable number of the MC realization was decided based on the comparison of statistical moments for different numbers of MC realizations. We have evaluated the DFN realization numbers for 100, 200, and 400. We found that the realization number less than 100 will lead to highly fluctuated variances of head and velocities. The mean values are relatively stable. With the realization numbers up to 400, the overall trends of head and velocity variances were obvious, except for some variations along the selected profiles. In this study, the 500 DFN realizations should be sufficient to judge the behavior of variance distributions. The discussion was added in the revised manuscript. Thanks for the comment.

2)
C=0 is set at the outflow (L15, p9). Why? This is quite unusual.
**Response**:

The C=0 at the outflow could represent the scenario that a large water body connected to the fractured rock on the right hand side. In this study, we knew that the downstream boundary could influence the results in the simulation domain. Therefore, the selected observation points for comparison of concentration were in the center area of the simulation domain. With a sufficient distance from the outflow boundary, the effect of the outflow boundary types (i.e., Dirichlet or Neumann) on the simulation results can be insignificant. Additional description had been added in the revised manuscript.

3)
- Fig. 12. The uncertainty in h decreases with X which is understandable, why is the uncertainty in vx suddenly increasing close to the outflow. I would appreciate if you could improve your comment on that.

**Response**:
Thanks for the comment. The head variances at inflow and out flow boundaries are reduced to 0 because the fixed head values are specified at the boundaries. Here the fixed head boundary conditions are assumed to be deterministic. The head variances should be zero at the inflow and out flow boundaries. However, the x-velocity variances at boundaries relies on calculations of head gradients and fracture connections at boundaries. These two parameters are not deterministic for MC simulations. Additionally, the artificial cuts of boundaries might also influence the head gradients and fracture connections at inflow and out flow boundaries. The integrated uncertainties of head gradients and fracture connections, therefore, lead to the increase of the velocity uncertainty in x-direction. The nonstationarity near simulation boundaries is usually observed in cases with heterogeneous porous media when the boundary conditions are specified deterministically. The discussion was added in the revised manuscript.

**Suggestion:**
What about anisotropy in the equivalent hydraulic conductivity? It should be quite easy to estimate by changing the flow BC.
**Response**:
Yes, the anisotropy in the equivalent conductivity can be estimated by changing flow directions. Thanks for the suggestion.

This is an important issue to clarify the relationship between fracture orientations and the behavior of anisotropy in the equivalent hydraulic conductivity. Because of the limited length of the paper, we will include the discussion of the issue in our future study. Thanks again for the valuable suggestion.

**Reviewer #1**

1)
3.1 Transport model verification by using HYDROGEOCHEM model
'The boundary conditions along the boundaries parallel to the flow direction are specified to be no-flow boundary conditions, except for the cross-shaped fracture network case, where a slightly upward flow along the vertical fracture is introduced'
Please explain better the upward flow. Is it a constant head value of 9 m as shown by the figure?

**Response**:
Yes, the upward flow was created by a constant head assigned on the top boundary of the vertical fracture.
In the revised manuscript, we have added detailed description for the B.C. of the case with cross-shaped fracture network. Additionally, the Figures 3c and d was modified to make the concept clear.
Thanks for the suggestion.

2)
3.2 Transport model verification by using analytical solution
On the basis of which criteria did you choose the dispersivities in the HYDROGEOCHEM model and the analytical solution?

**Response**:
We proposed that the fractures have rough surfaces with numerous contact points and fractures consist of the void space enclosed between two impermeable surfaces, which in a topological sense constitutes a two-dimensional porous medium. Therefore, the concept of porous fracture plates was employed in this study to formulate three-dimensional DFNs (Pruess and Tsang, 1990). Such porous fracture plates enable the use of

HYDROGEOCHEM model and the analytical model for validations (please see page 2 in "Mathematical formulas and numerical models").

The isotropic dispersivity used for the test cases was 0.1m. This value was determined based on the study of scale-dependent dispersivity proposed by Gelhar et. al.(1992). In our study, the interested scale is approximately on the order of 1m. Therefore, in our validation cases with the domain sizes of 2m, we use 0.1m to be the isotropic dispersivity for the transport simulation. The discussion was modified. Please see page 8 line 18 for details. Thank you for the comment.

**References:**

Pruess, K., Tsang, Y.W., 1990. On two-phase relative permeability and capillary pressure of rough-walled rock fractures. Water Resour. Res., 26, 1915–1926.

Gelhar, L.W., Wetly, C., Rehfeldt, K.R., 1992. A critical review of data on field-scale dispersion in aquifers. Water Resour. Res., 28(7), 1955-1974.

3)
4.1 Transport model verification
'The longitudinal dispersion is relatively obvious as compared to the transverse dispersion'. Explain this sentence.

**Response**:

Thank you for the comment. We will modify the discussion in the manuscript. In the comparison cases, the isotropic dispersivity was assigned for the simulation. Based on the x-direction uniform flow applied to the model, the effect of directional dispersion on the transport should be the same. The dispersion coefficient for x- and y-directions are:

$$\boldsymbol{D}_{xx}(\mathbf{x}) = \alpha_L(\mathbf{x})\frac{v_x v_x}{\bar{v}} + \boldsymbol{D}_o(\mathbf{x}),$$

(1)

and

$$\boldsymbol{D}_{yy}(\mathbf{x}) = \alpha_T(\mathbf{x})\frac{v_x v_x}{\bar{v}} + \boldsymbol{D}_o(\mathbf{x}),$$

(2)

where $a_L(\mathbf{x})$ is the longitudinal dispersivity in the principal flow direction. $a_T(\mathbf{x})$ represents the transverse dispersivity, which is perpendicular to the longitudinal dispersivity. Notations $v_x = v_x(\mathbf{x})$ and $v_y = v_y(\mathbf{x})$ are the seepage velocities in different directions in the porous fractures and $\bar{v}$ represents the

magnitude of the seepage velocity. Notation $D_o(\mathbf{x})$ is the effective molecular diffusion coefficient.

In the test example, the contribution from the advective transport is mainly from the seepage velocity in the x-direction. The discussion was modified. Please see page 10, line 28 for details.

4)

Figure 9

Please define in a careful way the parameters $P_{21}$ and $P_{32}$ as well as $P^e_{21}$ $P^e_{32}$ $P^t_{32}$ as the notations are quite misleading.

How did you calculate those parameters and do they differentiate among each other? And also better interpret the graph on the basis of those parameters and others such as fracture hydraulic conductivity and equivalent hydraulic conductivity. The discussion of graph 9d is not clear. Please provide a more accurate interpretation of the results.

**Response**:

Thanks for the comments. The definitions for different fracture intensities were clearly defined in the revised manuscript. Additionally, the calculations of these values were presented with details.

In this study the parameters $P_{21}$ and $P_{32}$ are the length of fracture traces per unit area (2D domain) and the area of fractures per unit volume (3D domain), respectively. We add a superscription "e" to indicate the intensity of effective fractures. The effective fractures are those that neglect isolated fractures in a rock. The superscription "t" represents the total fracture intensity for a rock. The concept of the equivalent hydraulic conductivity is to support the discussion of up-scaling issue. The detailed information and discussion was provided in the revised manuscript. Please see page 12 for details.

5)

5 Conclusions

The Conclusion is a mere summary of the obtained results. Rewrite the conclusion adding a more extensive interpretation and discussion of the results, including clarifications on the novelty of the proposed approach and how it would provide a benefit to the scientific community.

**Response**:

Thanks for the comments. Yes, there are novelty of the proposed approach

and benefits to the scientific community.

Numerical solutions to the ADE based on the Eulerian approach have not been widely implemented to DFN. This is because of computational issues such as numerical dispersion and convergence in the model for complex fracture connections. With the developed model, we also found interesting results that can contribute to the research communities of flow and transport in fractured rocks. The conclusion was improved based on the suggestions. Thanks for the comment.

**Anonymous Referee #2**

This is a very interesting topic; the efforts of the authors should be applauded by the community. The only concern I have is the practical application of this innovation. I am wondering if the authors can find experimental data to compare with their numerical calculations?

**Response**:

Thanks for the positive comment. Laboratory experiments can provide data for simple fracture connections. There are existing gaps between DFN models and field verifications. For realistic sites, the main difficulty is the measurement technologies that can be used to quantify the detailed fracture distribution in a rock. The site-specific fracture distributions are typically obtained from scanline or window samplings applied to available outcrops. It is even more difficult to conduct flow and transport experiments at sites with realistic scale and complexity. Most field experiments might focus on the equivalent behavior of flow and transport in fractured rocks. Such results can be the observations for evaluating the concept of parameter upscaling in DFN models. We also discussed this issue in the study.

There have been many numerical models proposed for DFN flow or transport simulations (i.e., Trinchero et al., 2016; Berrone et al., 2018; Fourno et al., 2019). Because of the limited technologies in measuring detailed flow and transport in a rock, the fracture statistics for rocks plays an important role in bridging the understanding between fracture distributions and flow and transport mechanisms. One can employ the fracture statistics to evaluate the possible flow paths or contaminant transport in fractured rocks. In this study, we used Monte Carlo Simulation (MCS) to quantify the influence of input uncertainty (i.e., the fracture intensity and distributions) on the output uncertainty. The insufficient data from sites were represented by the uncertainty and can be used for risk assessments or engineering designs. The references and

discussion were added in the revised manuscript.

References:
Berrone, S., Fidelibus, C., Pieraccini, S., Scialò, S., Vicini, F., 2018. Unsteady advection-diffusion simulations in complex Discrete Fracture Networks with an optimization approach. Journal of Hydrology 566, 332-345.
Fourno, A., Ngo, T.-D., Noetinger, B., La Borderie, C., 2019. FraC: A new conforming mesh method for discrete fracture networks. Journal of Computational Physics 376, 713-732.
Trinchero, P., Painter, S., Ebrahimi, H., Koskinen, L., Molinero, J., Selroos, J.-O., 2016. Modelling radionuclide transport in fractured media with a dynamic update of Kd values. Computers & Geosciences 86, 55-63.

We believe that the revised manuscript has addressed all the concerns proposed by Editor and reviewers. Please feel free to contact me if you have further questions regarding to this submission. Thank you for your time.

Sincerely,

Chuen-Fa Ni, Ph.D.,
Professor at Graduate Institute of Applied Geology,
National Central University,
Taoyuan City, 32001, Taiwan.
Email: nichuenfa@geo.ncu.edu.tw
Tel: 886-3-4227151 ext. 65874